# Prolonged Disability following Re-Exposure after Complete Recovery from Aerotoxic Syndrome: A Case Report

**DOI:** 10.3390/ijerph20247156

**Published:** 2023-12-08

**Authors:** Race Creeden, Nick Blonien, Jill K. Schultz, Jessika Wheeler, Erik L. Haltson, Zeke J. McKinney

**Affiliations:** 1HealthPartners Occupational and Environmental Residency, HealthPartners West End Clinic, 1665 Utica Ave. S., Suite 100, St. Louis Park, MN 55416, USA or creed014@umn.edu (R.C.); or blon0035@umn.edu (N.B.); erik.l.haltson@healthpartners.com (E.L.H.); 2Division of Environmental Health Sciences, School of Public Health, University of Minnesota, 1260 Mayo Building, MMC 807, 420 Delaware St. SE, Minneapolis, MN 55455, USA; 3Bright Eyes Vision Clinic, 13889 Ridgedale Drive, Minnetonka, MN 55305, USA; drschultz@thechildrenseyedoctor.com; 4NovaCare Rehabilitation, 480 Highway 96 East Suite 120, Vadnais Heights, MN 55127, USA; jesmwheeler@novacare.com; 5HealthPartners Institute, 8170 33rd Ave. S., Minneapolis, MN 55425, USA

**Keywords:** aerotoxic syndrome, organophosphate, tricresyl phosphate, neurotoxicity, aerospace medicine, fume event, acute brain injury

## Abstract

Aerotoxic syndrome is a rarely recognized and poorly defined illness resulting from exposure to products of oil combustion in commercial aircraft bleed air systems. The pathophysiology of this syndrome is not well understood. A 42-year-old male pilot developed duty-limiting symptoms consistent with aerotoxic syndrome following exposure to airline cabin fumes and recurrence of symptoms following re-exposure despite apparent full recovery from the initial exposure event. Given his persistent symptoms and concern for immediate debilitation/incapacitation if re-exposed, the pilot was deemed medically unqualified for return to commercial piloting duties and a disability determination was made. Consideration of aerotoxic syndrome in aircrew members with unusual or episodic symptoms of neuro/cognitive dysfunction is essential for further characterizing this occupational illness, guiding medical management, and understanding its impact on aviation workforce health.

## 1. Introduction

Aerotoxic syndrome is a hypothesized constellation of symptoms occurring secondary to exposure to toxic vapors produced by the bleed air system of commercial aircraft. The frequency at which this occurs is thought to be as high as 1 in every 66 flights, or as low as 1 in every 2000 flights [1]. Acute symptoms generally occur within 48 h of exposure to the toxic fumes and can include skin, eye, and respiratory irritation, headaches, nausea, and vomiting. Some go on to develop chronic symptoms of cognitive impairment, gastrointestinal distress, fatigue, and muscle aches [2]. Given some uncertainty regarding the pathophysiology of this condition and its variable degree of symptomatology, recognizing this syndrome as a defined clinical entity and assigning medicolegal causation has been contentious [1]. Airline and aircraft manufacturers have disputed that such symptoms could result from exposure to aircraft bleed air [1].

The syndrome at present is thought to be caused by engine oil lubricants, particularly the organophosphate compound tricresyl phosphate (TCP) after being heated to high temperatures in the engine, with vapors in bleed air that supplies ventilation to the aircraft cabin and flight deck [1]. Tricresyl phosphate has been shown to have multiple negative toxic effects in humans, especially from acute and high-dose exposures [3]. The toxicity of long-term exposure is less clear, but is thought to include neurologic insult resulting in weakness, tremors, and even paralysis [3]. The exact mechanism through which TCP produces these effects is not clear, but the generation of autoantibodies that damage neural tissue is thought to play a role and may explain the disability seen in some aircrew members who have experienced chronic low-dose exposure [4,5,6]. More recent literature has compiled the multiple proposed mechanisms of toxic injury, which not only include neural autoantibodies but also the increased toxicity of ultrafine particles, the potential for individual susceptibility to these exposures, the underestimation of toxicity for mixtures of compounds at low levels as compared to that of each individual compound, the distinct variation in toxicity for chronic low-level exposures versus acute higher-dose exposures, and the potential for endocrine disruption [7].

The patient described in this case report exhibited symptoms consistent with aerotoxic syndrome. This case report demonstrates the evaluation and management of this patient, as well as his disposition and continued care. This case is an excellent example of the importance of considering aerotoxic syndrome as a diagnosis in aircrew after a "fume event", not only by occupational medicine physicians, but by physicians who may have first contact with the patient as well. While occupational and aerospace medicine physicians should certainly be familiar with this syndrome, emergency and primary care providers may also encounter workers presenting with acute symptoms after initial exposure, as they often serve as the first point of contact after a work-related injury. Early recognition of this syndrome may help minimize risk of re-exposure and establish a well-informed plan for the patient’s return to work.

## 2. Detailed Case Description

### 2.1. Mechanism of Injury and Initial Presentation

A 42-year-old Airbus A320 pilot for a prominent airline company, with 14,000 h of flight time over 20 years, approximately half of which was in the A320, had just landed from a cross-country flight in Florida. Because of the hot weather, the air conditioner on the flight deck was switched on for 60–90 s, during which the pilot immediately observed a “dirty sock” smell. He promptly shut the air conditioner off. Approximately 20–30 min later, he was directly observed by his supervisor, who noticed he was having difficulty constructing coherent speech. Around the same time, his girlfriend also noticed similar deficits while on a video call with him. The pilot reported feeling “intoxicated.” Emergency medical response was activated, and he was transported by ambulance to a local emergency department for evaluation.

In the emergency department, he continued to describe feelings of intoxication, dizziness, blurry vision, headache, and anxiety. Objective findings at that time included slurred speech, tremor in his hands, and swollen cervical lymph nodes. These signs and symptoms resolved after about two hours in the emergency department, and he was released without further workup or treatment. Two days later, he was seen by his primary care physician, reporting persistent mild dizziness and word-finding difficulty. His blood carboxyhemoglobin level was analyzed and found to be within the normal range, though this did not exclude CO intoxication given it that was two days after the exposure. He was then referred to hyperbaric medicine for consideration of hyperbaric oxygen therapy (HBOT), although this was not administered given the absence of a clinically recognized indication for treatment.

He was then referred to occupational medicine for toxicologic consultation. At the initial appointment 10 days after the exposure, he reported increased fatigue, anxiety, photosensitivity, dizziness, short-term memory impairment, and difficulty spelling and driving. He did note that his symptoms had been gradually improving since the initial exposure.

### 2.2. Evaluation and Management

During this initial visit in the occupational medicine clinic, the patient was noted to have symptoms similar to an acute brain injury, such as difficulty concentrating, fatigue, photosensitivity, and short-term memory impairment. Convergence insufficiency was identified on physical exam. The Sport Concussion Assessment Tool 3 (SCAT3) symptom evaluation portion was used during the initial appointment and subsequent encounters as a means to objectively measure and quantify symptom severity [8]. His scores on this assessment over time are noted in Figure 1, and a full listing of all scores over time is available in Appendix A. At the conclusion of the initial visit, he was prescribed a standard postconcussion syndrome protocol. This included minimizing screen time, bright light, and loud noise exposure, strenuous activity, and caffeine, alcohol, and tobacco use. He was advised to use over-the-counter analgesics such as Tylenol or ibuprofen as needed. A referral to optometry for evaluation of vision and convergence abnormalities was placed, as was referral to physical and occupational therapy for vestibulo-ocular, balance, and cognitive rehabilitation. Because of the symptoms he described and their expected interference with safe aircraft operation, he was removed from the workplace pending resolution of symptoms.

At one-month post-exposure, he continued to report impairments in short-term memory and word finding, difficulty concentrating, headaches, fatigue, clumsiness, poor depth perception, photophobia, and peripheral vision disturbance, though he did note these symptoms had slightly improved since the previous visit. At the initial optometry evaluation, he was noted to have severely reduced near point of convergence, poor distance fusion resulting in diplopia at far distances, inaccurate saccade, pursuit eye movements, reduced accommodative ranges for age, and abnormal egocentric localization. The patient experienced increased dizziness and eye strain when testing horizontal and vertical visual ocular reflex and visual motion sensitivity. Automated visual field testing was performed and revealed reliable indices and generalized constriction of the right eye and no significant defects in the left visual field. He was prescribed therapeutic prism lenses, and in-office neuro-optometric rehabilitation was recommended. At the initial combined occupational and physical therapy appointment, it was noted that the frequent headaches were the patient’s main complaint, along with visual deficits noted on the SCAT3. As the patient was already referred for neuro-optometric rehabilitation, formal occupational therapy was deferred. At least 12 sessions of skilled physical therapy were recommended to address these symptoms. He continued treatment with physical therapy until 4 months post-exposure, at which point it was felt he had sufficient improvement to warrant a transition to a home exercise program. Some visual deficits remained, and he started formal neuro-optometric rehabilitation at six months post-exposure, which continued over a period of six months for a total of 33 visits, though visits were scattered due to disruptions in service availability during the COVID-19 pandemic. During the second half of his course of vision therapy, he noted that the use of a virtual reality gaming headset seemed to help his vision deficits significantly.

Approximately ten months after the initial exposure, the patient’s Federal Aviation Administration (FAA) medical examiner reported to the treating occupational medicine physician that a formal neurocognitive evaluation would be required before medical certification to fly could be reconsidered. By this point, the patient was not experiencing any subjective cognitive impairment, and stated that he was able to play a racing video game for over an hour with no fatigue and was scoring well on an informal cognitive training (“brain game”) application on his phone. Formal neuropsychological and neurocognitive testing was completed six months after exposure and was normal, with scores above average in all tested domains. The FAA also required that the patient be evaluated by neurology and specifically neuro-ophthalmology (as opposed to general ophthalmology or optometry) to confirm normal neurological status and resolution of visual field deficits, respectively. Testing confirmed normal neurologic and visual function, and these results were released to the FAA.

At nine months post-exposure, the patient confirmed that he felt he was back to his pre-exposure baseline. All planned evaluations and treatments were complete. After some delay in acquiring medical records of his immediate post-exposure care in Florida, he applied for medical reinstatement with the FAA 13 months after exposure. He received medical clearance to return to flight by the FAA 16 months after exposure and completed simulator-only training for four months after that point with no difficulty.

Twenty months after exposure, he resumed flying and had a total of four uneventful flights in an Airbus A320 over a period of one week before another symptomatic episode occurred. On the morning before the last of these flights, he woke with double vision and a headache, which resolved after showering and drinking coffee. He flew from the United States (US) to an airport in Mexico later that day and upon landing began experiencing difficulty focusing vision, rapid heart rate, hand tremor, and headache. He had not noted any unusual odor during any of these four flights. He was directed to the local hospital for emergency evaluation. The physical exam at that time was notable, showing an elevated blood pressure of 152/96 mmHg, hand tremor, diplopia, blurred vision, and visual field loss in the right upper quadrant of the right eye as determined by confrontation visual field testing. Laboratory testing, electrocardiogram, and head CT scan were normal. The evaluating physician noted that the patient was not fit to fly and recommended admission to the hospital for further assessment. He instead chose to return to the US as a passenger on a commercial flight the next day. He was seen by his occupational medicine physician about two weeks after this incident and reported persistent mild hand tremor bilaterally, intermittent right-sided frontal headaches, a non-limiting sense of imbalance, vision abnormalities, short-term memory problems, word-finding difficulty, and fatigue. He was deemed unfit to return to flying indefinitely, and it was anticipated that he may be permanently disqualified from piloting commercial aircraft given concern for re-exposure and recurrence of at least transiently debilitating symptoms.

The patient restarted neuro-optometric rehabilitation as he had noted a more frequent need for corrective lens use following his second syndromic episode. He was also referred for formal occupational therapy focusing on memory and cognitive rehabilitation, as he reported significant issues with reading and multi-step problem solving. He noted that insufficient sleep and cognitively challenging tasks seemed to exacerbate symptoms of double vision, headache, and fatigue. He also reported excellent physical exertional tolerance, as demonstrated by the absence of symptom exacerbation with jiu jitsu training five days per week. Per his occupational therapy evaluation and treatment, his motor coordination returned more quickly after this second exposure event, but he nevertheless had difficulty with completing simple and familiar motor coordination tasks while doing simple cognitive loading tasks such as memory games and word generation, demonstrating persistent difficulty with divided attention and multitasking.

### 2.3. Outcome and Disability

As of the patient’s most recent clinical evaluation at slightly more than three years after his initial exposure, the patient’s symptoms are mild, stable, and significantly improved from the acute post-exposure period. Vision and occupational therapy are helpful but so far have been unsuccessful in resolving his symptoms. He continues to experience visual impairment (convergence insufficiency, blurry vision, double vision), fatigue, and short-term memory deficits, which all seem to worsen with cognitive challenge and poor sleep.

The patient currently remains unfit for return to commercial piloting duties given the concern for vapor re-exposure and symptom recurrence. It is also suspected that his persistent mild symptoms may get acutely worse while under significant cognitive load as a working pilot, at least at the present time. It was discussed that it may be reasonable for him to return to private piloting or travel as a passenger on a commercial aircraft while wearing a respirator with an organic vapor cartridge. However, it is likely that he will be permanently disqualified from aviation duties where the safety of others is under his direct purview (i.e., working as a commercial pilot). He has not returned to professional aviation at this time and is currently pursuing other career options. He is approaching but has not yet reached maximum medical improvement (MMI), and consequently, a permanent partial disability rating has not yet been determined.

## 3. Discussion

Civilian aircraft adopted the practice of using unfiltered air from the compressors of the aircraft engines (bleed air) to supply cabin ventilation in the late 1950s, and it was noted soon after that crew members began to have adverse health effects. This was postulated to result from exposure to engine oil components leaking into the compressor air [9]. It is also well established that several toxic substances are produced from thermal decomposition when engine oil is exposed to high temperatures in the absence of oxygen, resulting in pyrolysis rather than combustion. These include ester-based lubricants, triaryl phosphates, organophosphate anti-wear additives, and amine antioxidants [9]. The most concerning of these is tricresyl phosphate (TCP), which is an organophosphate substance used as a flame retardant and lubricant for oils exposed to extreme pressures [10]. Interestingly, the Airbus A320, which was the type of aircraft piloted by this patient, has historically had challenges with cabin air quality [11].

Organophosphates such as TCP are toxic at high doses over short periods of time and can cause nausea, vomiting, diarrhea, swollen parotid glands, myalgia, cramping, paresthesia, tremors, and symmetrical polyneuritis [10]. Interestingly, the patient discussed in this case did have neck lymphadenopathy after his first exposure, which may have been parotid gland swelling mistaken for lymphadenopathy. There have been several recorded outbreaks of illness caused by acute ingestion of TCP throughout history. During prohibition, a drink called “ginger jake”, which required the use of TCP to extract Jamaica root, poisoned 10,000–15,000 people. In Europe in the 1930s and 1940s, TCP poisoned hundreds of women when it was used in abortifacients and at least 80 members of the Swiss Army when it was used by mistake as cooking oil [12].

Although it is understood that acute exposure can lead to neurotoxicity, chronic exposure to lower levels of TCP has not been as extensively studied, and current research is ongoing to further delineate symptom patterns and biomarkers that can signal long-term exposure and subsequent disease [4,13]. There is a growing concern that neurotoxicity caused by long-term exposure may be attributable to autoantibodies attacking endogenous neural proteins [4,5,6]. This may be similar to what occurs after a concussion or traumatic brain injury, as autoantibodies have been hypothesized to play a role in the disease process for this condition as well [14,15]. Repeated organophosphate exposure may also cause neurologic harm through other mechanisms, such as changes in gene expression, increases in oxidative stress due to free radical formation, neuroinflammation due to proinflammatory cytokine production, endocrine disruption, and impaired axonal transport through alterations in motor proteins and the neuronal cytoskeleton [7,13]. Another area of uncertainty is the lack of reported symptoms among passengers of commercial airliners as compared to aircrew. It is hypothesized that because passengers have less frequent exposure to TCP, this would explain the lack of symptoms and lower susceptibility to neurotoxic damage, though in some cases, biomarkers of exposure have been found in passengers even without symptoms [9,16]. Generally, because the neurotoxicity of TCP exposure is thought to be caused at the cellular level, there are no current imaging techniques to aid in the diagnosis of acute brain injury secondary to inhalational exposure, though research is being carried out in that area as well [17]. In the context of the multiple proposed mechanisms of TCP or mixed compounds containing TCP leading to aerotoxic syndrome, more contemporary literature has proposed standard protocols for documenting and evaluating aerotoxic syndrome events when they occur [7]. However, given the extremely contentious nature of this condition both in the medical and legal spheres, efforts should be made to implement standardized pre-employment examinations, with particular attention to baseline biomarkers [1]. For example, it is recommended that after an exposure event, blood tests for cholinesterase, carboxyhemoglobin, neuropathy target esterase, volatile organic compounds, etc., in addition to urine testing for organophosphates should be conducted, but these will have more utility in demonstrating adverse physiologic effects if a pre-exposure baseline is available [7]. It is also important to note that while much attention has been given to TCP so far, there are many other chemical components of engine oil and hydraulic fluid that are thermally degraded and introduced into the cabin ventilation, such as phenol isopropyl phosphate, amines, carboxylic acids, carbon monoxide, and hundreds of different volatile organic compounds, which also may contribute to the presentation seen in aerotoxic syndrome [7]. 

Regarding disability and prognosis for patients with aerotoxic syndrome, there is a paucity of information in the medical literature. Several publications note that symptoms can last 1–2 weeks or considerably longer in some patients, and can even be permanent in others [2,7,9,18]. In this regard, aerotoxic syndrome seems to be similar to postconcussion syndrome in that 57–95% of those with this syndrome fully recover within 1–2 weeks, but there is a subset of the population that may not recover for months or years [19,20,21,22,23,24]. It has also been established that the severity of the initial injury does not necessarily correlate with the severity of postconcussion symptoms, and that recovery is highly dependent on individual factors like access to rehabilitation services, social and environmental factors, and baseline neurocognitive functioning [20,22,24]. Re-exposure to TCP can also cause recurrence of symptoms, as was seen in this case, and this is often associated with increased disability, with some pilots being permanently disabled. One study of 274 pilots who had suspected TCP exposure events found that 44% reported symptoms lasting for days to weeks after exposure, 32% reported symptoms lasting for weeks to months after exposure, and 13% experienced chronic ill health leading to loss of flight status [9]. More recently, virtual reality has been promoted as a potential tool to help recovery from traumatic brain injury, and the patient discussed in this case used a virtual reality headset, with moderate improvement seen in his visual and balance deficits [25]. In addition, video games and phone-based applications have the potential to serve as a useful adjunct to traumatic brain injury rehabilitation and thus may be beneficial for rehabilitation in aerotoxic syndrome [26,27]. This patient was able to use both of these well and noted a modest subjective benefit. As seen in this case, multidisciplinary treatment is almost always necessary with postconcussion syndrome in order to optimize health outcomes. Therefore, a holistic and comprehensive approach to managing a patient with aerotoxic syndrome may also involve neurology, psychiatry, psychology, optometry/ophthalmology, and occupational/physical therapy, among other specialty services [24]. In addition to a multidisciplinary approach, a combination of pharmacologic and nonpharmacologic treatment has proven beneficial for patients with traumatic brain injury, which may apply to aerotoxic syndrome as well [22,28,29]. Future work needs to address the prevalence of affected individuals in terms of whether or not a return to their duties has been or can be achieved, given that re-exposure is anticipated to incite symptoms even after resolution.

## 4. Conclusions

Aerotoxic syndrome is an enigmatic, rarely diagnosed, and under-recognized constellation of symptoms thought to occur from inhalational exposure to tricresyl phosphate, which can occur in air crew when this organophosphate leaks into aircraft cabins from engine bleed air systems. Aircrew with aerotoxic syndrome may experience a wide array of acute and chronic symptoms affecting many different organ systems. Clinicians should have a high degree of suspicion for aerotoxic syndrome when treating commercial aircrew for symptoms that may resemble the presentation of traumatic brain injury, especially if there was known exposure to contaminated cabin ventilation. In addition, clinicians should consider treatment similar to that of traumatic brain injury, engage with appropriate specialists early, and remain open to unconventional or emerging treatment modalities such as virtual reality or mobile traumatic brain injury recovery tools.

## Figures and Tables

**Figure 1 ijerph-20-07156-f001:**
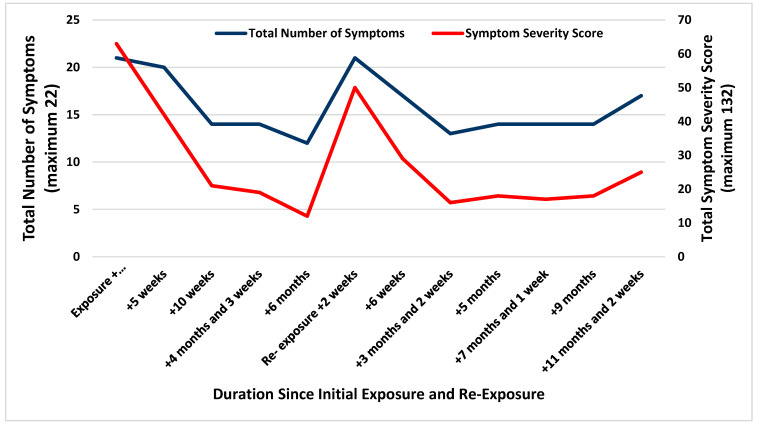
SCAT3 number of symptoms and severity score following initial exposure and re-exposure.

## Data Availability

Data are contained within the article and Appendix A.

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
