# Peer review of "Prolonged Disability following Re-Exposure after Complete Recovery from Aerotoxic Syndrome: A Case Report"

_ijerph, 2023, doi:10.3390/ijerph20247156_

Round 1

Reviewer 1 Report

Comments and Suggestions for Authors

This is a well-written case report of a 42-year-old pilot with chronic symptoms consistent with aerotoxic syndrome. Incoherent speech, dizziness, headache, blurred vision, and tremor of the hands were initial symptoms. Later he reported short-term memory impairment, word-finding difficulties, impaired concentration, headaches, fatigue and visual impairment. Visual impairment, optometric investigation and "unconventional" rehabilitation activities are well described. I have some minor remarks: 

page 2, line 46 and page 7, line 294: there is much focus on TCP, however other organophosphates, volatile organic compounds (mainly solvents) and carbon monoxide are probably responsible

page 2, line 75: Flying history of this pilot should be reported. How many years or flying hours is this pilot flying an Airbus 320?

page 2, line 89: a normal COHb level, 2-3 days after exposure does not exclude CO intoxication

page 4, line 166: Is the term amaurosis correct for a field defect in the right upper quadrant?

page 6, line 273 the percentage 433 is not correct

References: ref 7: please delete Susan, Jonathan, Vyvyan; S, J, V

Ref 12: no capitals

Author Response

  1. Thank you to the reviewer for their thorough review of our manuscript and for their thoughtful comments! We have addressed each of the reviewer's considerations below with enumeration of these points for ease of comparison:

    1. We added an additional sentence to the end of paragraph 3 in the discussion: "It is also important to note that while much attention has been given to TCP so far, there are many other chemical components of engine oil and hydraulic fluid that are thermally degraded and introduced into the cabin ventilation, such as phenol isopropyl phosphate, amines, carboxylic acids, carbon monoxide and hundreds of different volatile organic compounds, which also may contribute to the presentation seen in aerotoxic syndrome.[7]"

    2. Flying history and hours were added. First paragraph under "Detailed case description". 

    3. Made an addition about carboxyhemoglobin level being normal 2 days after exposure not ruling out CO intoxication. 

    4. Changed to "visual field loss in the right upper quadrant of the right eye as determined by confrontation visual field testing." The term amaurosis came from the Mexican hospital ER notes which were in Spanish, and which we reproduced for the purposes of this being the specific verbiage used by the evaluating physician.

    5. We have reviewed the reference Michaelis et al. (2017), in which it was stated, "The chronic cohort (13%) reported ill health at 37-433% above the controls.", which is the source from which the 433% was derived. However, in reviewing that article further, it is difficult to interpret from where that percentage was precisely derived, i.e., it may be from specific data that is not included in that publication. So while presumably that percentage is correct, e.g., 100 controls versus 533 cases, we opted to remove reproduction of this number based on this reviewer's comment for clarity. We consequently modified this sentence to read, "One study of 274 pilots who had suspected TCP exposure events found that 44% reported symptoms lasting for days to weeks after exposure, 32% reported symptoms lasting for weeks to months after exposure, and 13% experienced chronic ill health leading to loss of flight status." To convey the point that the time course for symptoms can vary widely.

    6. Corrected.

    7. Corrected. 

Reviewer 2 Report

Comments and Suggestions for Authors

This is a case report. The paper is extremely well written. 

My only suggestions is to remind that the aircraft engines in question are jet engines and the symptoms of Aerotoxic Syndrome were first recorded in the 1950's (as they have done) but in military aircraft, jet engines in civilian commercial aircraft came a little later.  It is also of note that there are no reports to my knowledge of Aerotoxic Syndrome in persons travelling on the Dreamliner Boeing 787 - it has a cabin air supply independent of the jet engines.

Author Response

Thank you for your comments! It will be interesting to see if there are any reports of aerotoxic syndrome in the 787 in the coming years, because if so that would seem to indicate that there are other factors at play besides bleed air, or possibly a sporadic mechanical malfunction that somehow enabled engine oil/lubricant biproducts to enter the cabin. Our understanding is that the 787 is set up this way because it is more efficient, so our hope is that eventually as efficiency standards increase the new aircraft entering production will all have this feature and maybe aerotoxic syndrome (despite it being clinically interesting) will be a thing of the past.

Reviewer 3 Report

Comments and Suggestions for Authors

This is a well written medical case report. The description of the clinical picture is excellent. It is very typical of the history and clinical signs of many cases of aerotoxic syndrome seen by other clinicians. The topic is currently very relevant.

The supporting introduction and discussion is generally well balanced and referenced. The comparison of the symptoms of aerotoxic syndrome with post-concussion syndrome is well made, though for my liking it is somewhat over emphasised. I would have liked to see more emphasis on the known possible mechanisms of pathogenesis of a chronic diffuse encephalopathy. The reference of Naughton & Terry (2018) is mentioned in the paper. However, the original paper the Terry (2012) published is not cited. In that review Terry went into a number of known mechanisms of harm from chronic low dose repeated exposures to OPs.    These included debilitation of both anterograde and retrograde axonal transport, the induction of neuroinflammation and the binding of OPs to certain amino acid residues (which has implications for protein misfolding pathologies). If these aspects could be introduced into the paper, to complement the, already present, discussion about neuro-antibodies, I think it would strengthen the paper and give the reader more insight into the possible pathogenic mechanisms in aerotoxic syndrome. 

Comments on the Quality of English Language

No comments, excellent English

Author Response

Thank you for your comments! We added the following to the third paragraph of the Discussion section to address this point: "Repeated organophosphate exposure may also cause neurologic harm through other mechanisms, such as changes in gene expression, increases in oxidative stress due to free radical formation, neuroinflammation due to proinflammatory cytokine production, endocrine disruption, and impaired axonal transport through the alteration of motor proteins and the neuronal cytoskeleton.[7,13]" It is worth noting that Budnik et al. (2023) [reference 7] have done a very good job (in our opinion) reviewing various proposed pathogenesis mechanisms, and given how recently that paper was published as well as due to the difference in the breadth of that paper versus our case report, we did not initially feel it was necessary to address this point in a great degree of detail. In general, we are trying to somewhat toe the line between keeping the paper accessable and easy to read versus going into great depth about pathogenesis (which the other authors we have cited do such a great job of detailing, probably much better than we could do in the context of this case report). If the reviewer believes this point requires more detail than modified here or if the reviewer has other suggestions, we are certainly willing to expand!